# Disruptive Behaviors in Physical Education: A Matched Study of Social Skills and Sport Practice in a Region of Spain

**DOI:** 10.3390/ijerph19031166

**Published:** 2022-01-21

**Authors:** Rubén Navarro-Patón, Marcos Mecías-Calvo, Raúl Eirín-Nemiña, Víctor Arufe-Giráldez

**Affiliations:** 1Facultad de Formación del Profesorado, Universidade de Santiago de Compostela, 27001 Lugo, Spain; ruben.navarro.paton@usc.es; 2Facultad de Ciencias de la Educación, Universidade de Santiago de Compostela, 15782 Santiago de Compostela, Spain; raul.eirin@usc.es; 3Facultad de Ciencias de la Educación, Universidad de A Coruña, 15008 A Coruña, Spain; varufe@udc.es

**Keywords:** relatedness, irresponsibility, fails to follow directions, distracts or disturbs others, poor self-management

## Abstract

Disruptive behaviors in physical education cause conflicts among students and, consequently, an abnormal development of classes. Therefore, finding the variables that can solve them is an urgent aspect to achieve an adequate learning environment in the 21st century school. The aim of this study was to analyze what happens to disruptive behaviors in relation to systematic and regulated sports and social practice in a sample of Spanish primary school students. Five hundred and forty-eight schoolchildren (276 were girls (50.4%)) participated with a mean age of 10.98 (SD = 0.71). The results show a significant main effect in terms of social skills in relatedness (*p* < 0.001), irresponsibility (*p* < 0.001), failure to follow directions (*p* < 0.001), distracting or disturbing others (*p* < 0.001), and in poor self-management (*p* < 0.001) with higher scores in disruptive behaviors in students with lower social skills. Regarding sports practice, only a significant main effect was found in relatedness (*p* < 0.001) and in poor self-management (*p* < 0.001), with the highest scores the schoolchildren who do not practice sports. Schoolchildren with high social skills obtain lower scores in disruptive behaviors. Likewise, schoolchildren who play sports have lower scores in relatedness and poor self-management.

## 1. Introduction

Disruptive behaviors in the educational field are generally understood as those that disturb, in some way, the teaching–learning process [1,2,3,4,5,6], and they are usually related to the transgression of the rules, considered a common behavior in the evolutionary development during childhood and adolescence that helps to strengthen the personality and social positioning [7]. 

These behaviors tend to occur in 10% of students in countries such as Norway or Japan, reaching up to 60% in countries such as Spain [8], affecting most schools and subjects causing a dysfunction of the educational process [9]. The area of physical education (PE), by its own idiosyncrasy, has been labeled as a curricular area where more inappropriate or disruptive behaviors occur [10,11,12]. This situation is cause for concern among teachers, who try to find the factors that influence or protect these behaviors to safeguard quality standards in teaching [13].

Among the most frequent disruptive behaviors that can be observed in the PE subject, we can find relatedness, irresponsibility, failure to follow directions, distracting or disturbing others, and poor self-management. When we talk about relatedness, it refers to behaviors that can be manifested through physical attacks (blows), verbal attacks (insults), and that can be carried out directly (against the person attacked) or indirectly through the dissemination of slander [14]. Irresponsibility refers to the behavior of the person who acts through excuses, blaming others for their behavior and way of acting [15]. Failure to follow directions refers to the omission of teachers’ instructions repeatedly and intentionally, interrupting the class untimely or without paying attention [16]. As in the previous case, distracting or disturbing others prevent the normal development of classes, affecting the rest of the students [17]. Finally, when we talk about poor self-management, we refer to the individual’s inability to inhibit uncontrolled expressions, teasing from classmates, anger, or difficulty in handling different situations that can occur in PE classes [18].

Scientific evidence reveals that most of the studies conducted on disruptive behaviors have been carried out with Secondary Education students [19,20] and, to a lesser extent, in Primary Education [21], producing a progressive increase in these behaviors with age, from 10 to 16 years [20,21]. Regarding gender, different studies indicate that boys have a higher disruptive behavior profile than girls [7,22,23,24], although contradictory results have also been found, indicating that girls in Secondary Education perform a greater number of disruptive behaviors than boys [21].

Although the causes and origins of these behaviors can be multiple (family, social, individual, curricular factors...), it is known that there are protective factors; understanding these as those that increase the individual’s ability to successfully face problematic situations, among which are social skills [25], as the set of specific and essential capacities to interact and relate effectively with others [26]. These social skills must be developed in a context that facilitates personal interaction and respect [27,28]. This context can be the systematic practice of physical activity and sport since it offers children and adolescents an adequate space to enhance personal and social aspects (i.e., belonging to a group, cooperation, respect) [29] and allows reducing antisocial behaviors in adolescents [30].

To the authors’ knowledge, social skills and systematized sports practice have not been analyzed, to date, as protective factors of disruptive behaviors in PE classes in schoolchildren aged 10 to 12 years. Therefore, the aim of this study was to analyze what happens to disruptive behaviors in relation to systematic and regulated sports practice, and social skills in a sample of Spanish students in Primary Education, aged 10 to 12 years. Based on this objective, the following research questions were posed: (a) Do schoolchildren aged 10 to 12 have a high score in the disruptive behaviors studied? (b) Do schoolchildren who have the highest score in social skills have the lowest score in different disruptive behaviors? (c) Do schoolchildren who practice sports systematically have lower scores in disruptive behaviors than those who do not?

## 2. Materials and Methods

### 2.1. Participants

In this cross-sectional, observational, and descriptive study [31] with a non-probabilistic and convenience sample, depending on the subjects that could be accessed, a total of 548 schoolchildren participated (49.6% boys and 50.4% girls) from public centers in Galicia (Spain). The age range was 10 to 12 years (M = 10.98, SD = 0.71). The distribution by subjects was: 176 (32.1%) from fifth grade and 372 (67.9%) from sixth grade.

### 2.2. Measurement Tools

An ad hoc questionnaire was used to collect sociodemographic data: age (years in number); gender (boy/girl); grade (fifth/sixth of Primary Education); and extracurricular sports practice in a sports club (yes/no).

To collect data on disruptive behaviors, the Disruptive Behaviors in Physical Education Questionnaire (CCDEF) was used. It is a valid and reliable questionnaire consisting of 17 elements preceded by an introductory phrase “Think about your own behavior in the EF class and tell us your grade according to the following statements” [18]. The items are on a Likert scale from 1 to 5 (where 1 means “never”; 2 “almost never”; 3 “sometimes”; 4 “often”; and 5 “always”). The scale has 5 factors: relatedness (“Threatening other classmates”), irresponsibility (“I move slowly on purpose”), failure to follow directions (“I do not follow instructions”), distracting or disturbing others (“I leave the group during an activity”), and poor self-management (“I make fun of other colleagues”).

To collect data on social skills, the Scale of Appropriate Behavior in Physical Education and Sports (CAEFD) was used. It is a valid and reliable questionnaire consisting of 32 elements preceded by the statement “During physical education class ...” [32]. The items are of the Likert scale type from 1 to 5 (where 1 means “totally disagree”; 2 “somewhat disagree”; 3 is equal to “neutral”; 4 means “somewhat agree” and 5, “totally agree”). Although the scale has 5 factors, for this research only the one that refers to social ability (SA) (“Do you hear when someone talks to you?”) was used.

### 2.3. Procedure

In the first place, the heads of the participating educational centers were informed, as well as the physical education teachers. Subsequently, the parents of the schoolchildren were informed about the protocol and the object of study, obtaining the informed consent of the legal guardians of the minors in order to participate in the research. All participants, as well as their legal guardians, were informed of the objective of the study, the voluntary and confidential nature of the responses, and the management of the data, as well as their rights as participants of the same, by virtue of the Declaration of Helsinki [33]. This research was approved by the Ethics Committee of the national platform Educa (Code 72021).

The tools used to collect information on the study variables of this research were administered in the classroom by the researchers themselves, without the presence of the teacher.

### 2.4. Statistical Analysis

All statistical analyzes were performed with SPSS software (v.25, IBM Corporation, New York, NY, USA). The level of statistical significance was established at *p* < 0.050.

Demographic data were expressed in frequencies, for categorical variables, and in measures of central tendency (mean and standard deviation), for quantitative variables. To know the differences between the CCDEF variables, the level of social skills (low = 1–2.5; medium = 2.51–3.50; high = 3.51–5) and sports practice (yes/no), and a multiple analysis of covariance (MANCOVA) was performed, using the Bonferroni statistic to know the significance of the interaction between the variables. To determine the effect size, eta squared (η^2^) was used. Sex and age were used as covariates to control for the influence of possible confounders.

## 3. Results

Five hundred and forty-eight schoolchildren participated in this research, of which 272 were boys (49.6%) and 440 (80.3%) practiced sports systematically in a sports club. Besides, more than two thirds of the schoolchildren had high social skills (88.3%) (Table 1).

### Results of the Analysis Based on Social Skills and Sports Practice

The results of the multiple analysis of covariance (MANCOVA, Table 2) regarding relatedness indicate that there is a significant main effect in the social skills factor (F (2, 540) = 186.461; *p* < 0.001; η^2^ = 0.41) with higher scores in schoolchildren with low social skills. A significant main effect was also found in the sports practice factor (F (1, 540) = 68.194, *p* < 0.001, η^2^ = 0.11) being higher in those who do not practice sports. Statistically significant results were also found in the interaction of both factors (F (2, 540) = 48.154, *p* < 0.001, η^2^ = 0.15), being higher in schoolchildren with low social ability who practice sports.

The results regarding irresponsibility, as in the previous dimension, show that there is a significant main effect in the social skills factor (F (2, 540) = 74.548; *p* < 0.001; η^2^ = 0.21) being higher in schoolchildren with low social skills. Likewise, there are significant differences in the interaction of both factors (F (2, 540) = 16,512; *p* < 0.001; η^2^ = 0.06) being higher in schoolchildren with medium social skills and play sports. No statistical significance was found in the sports practice factor (*p* = 0.596).

Regarding failure to follow directions, in the same line as in the previous dimensions, the results indicate that there is a significant main effect in the social skills factor (F (2, 540) = 321.583; *p* < 0.001; η^2^ = 0.54) also being higher in schoolchildren with low social skills. Statistically significant differences were found in the interaction of both factors (F (2, 540) = 11.213; *p* < 0.001; η^2^ = 0.04) being higher in schoolchildren with medium social skills and who do not practice sports. No statistical significance was found in the sports practice factor (*p* = 0.610).

The results on distracts or disturbs others indicate that there is a significant main effect in the social skills factor (F (2, 540) = 161.721; *p* < 0.001; η^2^ = 0.37) being higher in schoolchildren who present low social skills. Statistically significant differences were found in the interaction between the social skill factor and sports practice (F (2, 540) = 46.823; *p* < 0.001; η^2^ = 0.15) being higher in schoolchildren who have medium social skills and do not practice sports. No statistically significant results were found in the sports practice factor (*p* = 0.065).

With regard poor self-management, the results show that there is a significant main effect in the social skills factor (F (2, 540) = 276.932; *p* < 0.001; η^2^ = 0.51) being higher in schoolchildren with low social skills. A significant main effect was also found in the sports practice factor (F (1, 540) = 59,716; *p* < 0.001; η^2^ = 0.100) being higher in schoolchildren who do not practice sports. Finally, statistically significant results were also found in the interaction of both factors (F (2, 540) = 73,014; *p* < 0.001; η^2^ = 0.21) being higher in schoolchildren who have low social skills and play sports.

## 4. Discussion

The aim of this study was to analyze what happens to disruptive behaviors in relation to systematic and regulated sports and social practice, in a sample of Spanish Primary Education students. Through this research, an attempt was made to respond to the behavior problems that occur in physical education classrooms [10,11,12,21], which, as mentioned, occur both in Primary Education [7,21] and in Secondary Education [19,20,21] coinciding with the end of childhood and adolescence [7].

These behaviors studied, and present in the sample that participated in this research, affect the functioning and context of the class, and, therefore, in the teaching–learning process [34].

The present study shows that several disruptive behaviors are more frequent when students’ social skills are low. In addition, it is necessary to emphasize that systematic sports practice outside the school context does not influence all the disruptive behaviors analyzed in the same way. The practice or not of sports does not influence behaviors such as irresponsibility, failure to follow directions, or distracting or disturbing others, which may be due to other factors such as demotivation or boredom [7], variables not studied in this research.

Regarding social skills, the degree of social ability on the part of students is inversely proportional to distracts or disturb others, as can be observed in our study. Students with higher scores in social skills present fewer acts of relatedness and poor self-management, variables that, in some way, are related to dissatisfaction with school, and that can motivate their dropout [35]. On the other hand, irresponsibility, low commitment, and failure to follow directions when carrying out the tasks proposed by teachers, or distracting or disturbing other colleagues in the execution of any type of task, may be motivated by the preference in the application of a traditional methodology by teachers [36], which can cause dissatisfaction and disturbance of the PE class environment [37].

Therefore, the answer to the research question posed: Are the schoolchildren who have the highest score in social skills the ones who give the lowest score to the different disruptive behaviors? The answer to this question is affirmative, since in all the dimensions of the disruptive behaviors studied, having high social skills implies lower scores in each and every one of the disruptive behaviors studied. For this reason, we want to place special emphasis on the importance of an initial and permanent training of teachers in the development of social skills and emotional education, necessary to guarantee that these aspects are worked with students and thus contribute to a lower production of disruptive behaviors.

In addition, teachers must maintain an understanding and empathic attitude promoting a teaching style that promotes student participation and leadership, fostering reciprocal responsibilities [36]. These aspects, mentioned above, are key to a better bond between students, the class and the teachers themselves [38,39].

The practice of sport or programmed, controlled, and systematized physical activity [40], and its context, allows the development of personal values such as social values (cooperation and respect), perseverance, or ability of exercise, especially in childhood and adolescence [41]. For these reasons, it would be expected that the scores for the disruptive behaviors studied in this research would be lower in schoolchildren who play sports than in those who do not. In this line, we must indicate that the results obtained in this study show that sports practice at these ages positively influences the scores for disruptive behaviors. 

All the dimensions studied achieved a lower score in schoolchildren who practice sports compared to those who do not, although only significantly in the dimension of relatedness and poor self-management. These results follow the line of the study by Pelegrín [42], which indicates that sports practice contributes to the prevention of antisocial behaviors, or to obtaining higher scores in mental strength, optimism, and perseverance in athletes compared to non-athletes [43], which allows reducing antisocial behaviors [30].

For all the above, what is the answer to the question of whether schoolchildren who practice sports systematically have lower scores in disruptive behaviors than those who do not practice it? The answer to this research question can only be answered affirmatively in the case of relatedness and poor self-management since the practice or not of sports does not influence the other dimensions of the disruptive behaviors studied (i.e., irresponsibility, failure to follow directions, distracting or disturbing others) data that correspond, in part, to previous studies [30,43,44,45]. Despite these results, it must be taken into account that sports practice develops bonds with their peers, awakening the feeling of belonging to a specific group or collective. The transmission of values through sport is reflected in compliance with the necessary regulations for sports development, learned, internalized, and respected by the participants [41], which would help reduce disruptive behaviors.

As limitations of this study, we must point out that the study was carried out by self-report of the students, with the implications that this has. On the other hand, the choice of educational centers, and therefore of the sample, was non-random. That is why the results of this research must be taken with caution and further research is required on the variables studied in this research (disruptive behaviors, social skills, and sports practice), as well as other territorial distributions in our country to check if these trends are repeated or are different.

## 5. Conclusions

As conclusions of this study, we must point out that Primary Education students who have high social skills attained lower scores in the disruptive behaviors studied (i.e., relatedness, irresponsibility, failure to follow directions, distracting or disturbing others, and poor self-management). 

On the other hand, students who do not practice sports scored higher in all disruptive behaviors studied, although only significantly in relatedness and poor self-management.

In the combination of social skills and sports practice, students who have low social skills and play sports obtained higher scores in relatedness and poor self-management; students who have medium social skills and do not play sports obtained higher scores in failure to follow directions and distracting or disturbing others, and finally, students who have medium social skills and play sports obtained higher scores in irresponsibility.

## Figures and Tables

**Table 1 ijerph-19-01166-t001:** Characterization of the sample.

Variables	Options	Frequency (%)
Age average (years)	10.98 (SD = 0.71)	-
Gender	Boy Girl	272 (49.6%) 276 (50.4%)
Sport practice	Yes No	440 (80.3%) 108 (19.7%)
Grade	Fifth of Primary Education Sixth of Primary Education	176 (32.1%) 372 (67.9%)
Social skills	High (3.51–5.00) Medium (2.51–3.50) Low (1.00–2.50)	484 (88.3%) 56 (10.2%) 8 (1.5%)

Note: Quantitative variables are expressed as mean and standard deviation, and qualitative variables are expressed as frequencies and percentage.

**Table 2 ijerph-19-01166-t002:** Results of disruptive behaviors based on social skills and sport practice.

Disruptive Behaviors	Sport Practice	High Social Skills		Medium Social Skills		Low Social Skills	
		Mean	SD	Mean	SD	Mean	SD
Relatedness	No	1.17	0.23	2.00	0.00	1.00	0.00
	Yes	1.17	0.31	1.83	0.33	3.00	0.00
	Total	1.17	0.30	1.89	0.28	2.00	1.06
Irresponsibility	No	1.73	0.58	2.75	0.00	2.50	0.00
	Yes	1.38	0.47	1.80	0.33	4.00	0.00
	Total	1.44	0.51	2.14	0.52	7.25	0.80
Failure to follow directions	No	1.60	0.47	3.25	0.00	1.25	0.00
	Yes	1.19	0.27	3.05	1.11	2.00	0.00
	Total	1.26	0.35	3.12	0.89	1.62	0.40
Distracting or disturbing others	No	1.33	0.36	3.00	0.00	1.25	0.00
	Yes	1.34	0.35	1.77	0.42	2.00	0.00
	Total	1.34	0.36	2.21	0.67	1.62	0.40
Poor self-management	No	1.30	0.41	3.00	0.00	1.00	0.00
	Yes	1.25	0.33	2.29	0.58	3.67	0.00
	Total	1.26	0.35	2.54	0.57	2.33	1.42

Note. SD = standard deviation.

## Data Availability

The data presented in this study are not available in accordance with Regulation (EU) of the European Parliament and of the Council 2016/679 of April 27, 2016 regarding the protection of natural persons with regard to the processing of personal data and the free circulation of these data (RGPD).

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
