# Peer review of "Disruptive Behaviors in Physical Education: A Matched Study of Social Skills and Sport Practice in a Region of Spain"

_ijerph, 2022, doi:10.3390/ijerph19031166_

Round 1
Reviewer 1 Report
First of all, I would like to congratulate the authors for this work. For me, as a teacher and scientist, this topic is very important and has a lot of value. I have enjoyed reading this manuscript very much.
This paper analyzes what happens to disruptive behaviors in relation to systematic and regulated sports and social practice in a sample of Spanish primary school students.
The manuscript fits well within the scope of the journal and the analysis is based on a good sample size. The methods are described well, but I have some vital recommendations on how to improve them. I will then write recommendations on how to improve it. I will present my recommendation at the end of this paper.
Title
The title is accurate and informative. I think it would be even more informative for readers if the region is mentioned. Suggested: Disruptive behavior in physical education: a matched study of
social skills and sports practice in a region of Spain.
Abstract
The abstract is clear, and complies with the general rules for writing a good abstract. It should not be modified. I consider this to be the most important section of the document since it will be read on many more occasions than even the paper itself.
Justification and theoretical framework
As I mentioned, I find this research extremely important. I do not disagree with the authors' justifications and read many very good arguments.
Methodology:
The methods section is good and well written. The sample is large enough to perform the selected methods. Nevertheless some recommendations are presented:
- Participants: We suggest including the sample characterization table present in the results section in this section, so as not to duplicate information.
- Instruments. I would like to read more about the validation and psychometric properties of the scales used (CCDEF and CAEFD). It is reported that they are factorial instruments, was a confirmatory factor analysis performed on both of them?
- Statistical analysis: It is recommended to report the justification of the statistical tests used. Data are presented as mean and SD. Did the data follow a normal distribution and was the Kolmogorov-Smirnov test applied to analyze this assumption? It is recommended to clarify this.
Results: They are all shown correctly and are easy to read for a person not used to interpreting statistical data.
Conclusions. They are clear and give an answer to the stated objectives. Another research that adds to the importance of the practice of sport and physical activity for the improvement of the physical and psychological health of students.
I recommend that this manuscript be sent for another round of review after minor revisions. I believe it should be published.
Reviewer 2 Report
1. In the manscriput, partial eta squared (η2) was used to determine the effect size. However, whether the sample size in the study is suitable for using partial eta squared, especially the sample size of low social skills is only 8?
2. Participants included different genders (boy and girl) and ages (10-12). However, it is not found in the results and discussions of the manscriput the match study of social skills and sport practice was not analysed and discussed based on gender and age.
Reviewer 3 Report
I am grateful for the provided opportunity to review the article.
The article presents significant and relevant results.
In general, I have only one very specific proposal that needs to be fulfilled while improving the quality of the article. It is necessary to provide a more exhaustive theoretical rationale and description of the key variables analysed in the study. This can be done in the introduction by expanding it or, even better, by providing a separate section “Theoretical rationale”.
The addition of this section to the article would enable a clearer understanding of the context of the research problem and its scientific exploration.
I would recommend that the conclusions are presented in more detail.
Round 2
Reviewer 2 Report
Thanks for the reply. No doubt.